# Liquid Biopsy and Multidisciplinary Treatment for Esophageal Cancer

**DOI:** 10.3390/cancers17020196

**Published:** 2025-01-09

**Authors:** Yuki Hoshi, Satoru Matsuda, Masashi Takeuchi, Hirofumi Kawakubo, Yuko Kitagawa

**Affiliations:** Department of Surgery, Keio University School of Medicine, Tokyo 160-8582, Japan

**Keywords:** esophageal cancer, liquid biopsy, ctDNA, circulating tumor cells, miRNA

## Abstract

Multidisciplinary treatment, including surgery, chemotherapy, or chemoradiotherapy, which are individualized based on the tumor progression, is used for esophageal cancer (EC) management. A novel testing modality for monitoring tumor burden throughout treatment is required to perform this individualized treatment. Liquid biopsy has recently become popular because conventional tests, such as upper gastrointestinal endoscopy and computed tomography (CT), are insufficient to evaluate minimal residual diseases. It requires taking a small body fluid sample and testing it for circulating tumor DNA (ctDNA), microRNA (miRNA), or circulating tumor cells (CTCs). Liquid biopsy may help predict EC prognosis and recurrence, stratify high-risk populations, and determine cases with complete responses to preoperative treatment. Additionally, it may determine the suitability for postoperative chemotherapy and conversion surgery. The potential of liquid biopsy to enhance treatment decisions will drive further EC treatment advancements.

## 1. Introduction

Esophageal cancer (EC) is the 11th most prevalent cancer and the seventh leading cause of cancer-related deaths in the world [1]. A total of 511,054 new EC cases were reported globally in 2022, with 445,391 deaths from the condition [1]. Adenocarcinoma, which primarily originates in Barrett’s esophagus, is the most prevalent histological type of EC in Western countries. In contrast, squamous cell carcinoma accounts for most cases in Asia [2]. The incidence and mortality have not significantly improved globally, and the prognosis remains poor [3].

Endoscopy, surgery, radiation therapy (RT), and chemotherapy are used for EC treatment. Endoscopic submucosal dissection is performed to manage early-stage EC, and surgery or chemoradiotherapy is considered when endoscopic resection is contraindicated [3]. Patients who are eligible for surgery receive neoadjuvant chemotherapy (NAC) and chemoradiotherapy (NACRT), followed by postoperative adjuvant chemotherapy based on their status (Figure 1). Unresectable EC is administered with systemic chemotherapy. Multidisciplinary treatment for EC has been individualized according to the tumor progression. Various testing modalities, such as computed tomography (CT), endoscopy, and positron emission tomography (PET), have been used to evaluate the tumor burden during treatment. However, their accuracy remains unsatisfactory [4,5]. Recently, liquid biopsy has become popular as a new diagnostic technique, and its importance is increasing [6]. This review investigates the progress and present status of various treatments, particularly surgery and perioperative therapy as well as addresses liquid biopsy, a new testing modality, and its potential use in multidisciplinary EC treatment.

## 2. Multidisciplinary Treatment for Resectable EC

### 2.1. Surgery

The first successful thoracic EC surgery was conducted by Torek [7]. Three-field lymph node dissection (3FD), including cervical and superior mediastinal dissection, has been recognized as an option for thoracic EC treatment since the 1980s [8]. The development of minimally invasive surgery was initiated after Cuschieri reported the world’s first thoracoscopic esophagectomy [9]. Takeuchi released the results of JCOG1409, a phase 3 trial that compared thoracoscopic esophagectomy with open esophagectomy for thoracic EC [10]. The data indicated noninferiority in overall survival (OS) (hazard ratio [HR]: 0.64, 98.8% confidence interval [CI]: 0.34–1.21, one-sided *p* = 0.000726). Robot-assisted esophagectomy has swiftly grown in popularity globally since Bodner described the first case of esophagectomy with da Vinci [11]. Chao demonstrated the effectiveness of robot-assisted esophagectomy in the REVATE trial, and further results are likely in the future [12].

### 2.2. Perioperative Treatment

INT113 and OEO2 trials first exhibited the efficacy of NAC [13,14,15]. The JCOG9204 and the JCOG9907 trials in Japan established preoperative CF (cisplatin +5-fluorouracil [5-FU]) therapy plus surgery as the standard treatment for esophageal squamous cell carcinoma (ESCC) [16,17]. NAC for esophageal adenocarcinoma (EAC) and esophagogastric junction carcinoma has been extensively established in Western countries. Preoperative ECF (epirubicin + cisplatin + 5-FU) therapy became the standard treatment for EAC in the United States (US) and Europe after the MAGIC trial [18]. ECF was subsequently replaced with FLOT (docetaxel + oxaliplatin + 5-FU/leucovorin) based on the FLOT-4 trial [19].

NACRT was developed concurrently with NAC in Western societies. The CROSS trial was the first randomized controlled trial of NACRT for EC. This trial revealed that the NACRT group outperformed the surgery-alone group in terms of the pathological complete response (pCR) rate and OS, and the CROSS regimen (carboplatin + paclitaxel + 41.4 Gy) was determined as the optimal therapy for NACRT [20].

POET [21], Neo-AEGIS [22], and ESOPEC [23] are trials that compared NAC with NACRT. In particular, the ESOPEC trial, which compared the FLOT and the CROSS regimen, revealed that the FLOT group demonstrated a better 3-year OS than the CROSS group, indicating that NAC may be more beneficial than NACRT [23]. These were all trials on EAC, whereas the JCOG1109 trial in Japan compared CF, DCF (docetaxel + cisplatin + 5-FU), and CF + radiotherapy (CF + RT) as NAC for ESCC, revealing 5-year survival rates of 51.9%, 65.1%, and 60.2%, respectively [24]. The result indicated that neoadjuvant DCF has overtaken CF as the standard treatment in Japan. The JCOG1109 and ESOPEC results are considered a significant step toward conventional therapy in the US and Europe, where NACRT is predominantly used. These may cause a paradigm shift in the future, with chemotherapy administered first and RT reserved for recurrence and other situations.

The CheckMate-577 trial investigated the efficacy and safety of nivolumab as an adjuvant therapy in individuals who underwent an esophagectomy after NACRT [25]. The nivolumab group demonstrated better disease-free survival in this trial; thus, adjuvant nivolumab became the norm in Western countries. The JCOG2206 study in Japan is being conducted to evaluate the add-on efficacy of nivolumab and S-1 in patients with ESCC who underwent esophagectomy following NAC but did not achieve pCR [26]. Furthermore, several other trials on perioperative treatment are currently ongoing globally, namely JCOG1804E, which explores the effect of nivolumab on neoadjuvant CF, DCF, and FLOT, with many more pivotal studies anticipated to appear [27,28,29].

## 3. Liquid Biopsy

Several perioperative treatments have been developed for resectable EC, and their efficacy has been identified via upper gastrointestinal endoscopy or computed tomography (CT). However, the accuracy and precision of these conventional modalities are unsatisfactory, particularly in diagnosing minimal residual disease (MRD). Kermani revealed that endoscopic observation after preoperative chemoradiotherapy for EC exhibited positive and negative predictive values for pCR of 64.2% and 75%, respectively, and concluded that this approach was insufficient for evaluating treatment efficacy [4]. Van Rossum’s systematic review indicated that the sensitivity and specificity of endoscopic biopsies after NACRT were 34.5% (95% CI: 26.0–44.1%) and 91.0% (95% CI: 85.6–94.5%), respectively, making the use of endoscopy to predict pCR questionable [30]. Similar studies have been conducted with CT with 33–55% sensitivity and 50–71% specificity for assessing tumor response after NACRT [31,32]. Furthermore, Alfieri utilized CT to evaluate changes in tumor volume, with sensitivity and specificity of 56% and 93%, respectively, indicating its limited role [5]. The need for a new technology to replace the predominantly used endoscopy and CT has been developed, and liquid biopsy has drawn attention. Liquid biopsy is a technique to collect and evaluate tumor-related substances from body fluids, including blood and urine [33]. It is less invasive than traditional tissue biopsy and is repeated several times, enabling tumor status evaluation over time.

The history of liquid biopsy began in 1869 when Ashworth discovered circulating tumor cells (CTC) in the blood of corpses [34]. Mandel and Metais discovered cell-free DNA (cfDNA) in blood in 1948, and Stroun revealed circulating tumor DNA (ctDNA) in 1989 [35,36]. Alix-Panabières and Pantel coined the phrase “liquid biopsy” in 2010 to describe the methodology for CTC [37]. However, the term has appeared to refer to all tumor-derived compounds in the body fluid, including ctDNA, microRNA (miRNA), and CTC. In recent years, analyzing these components in patients with EC has enabled a more accurate treatment efficacy evaluation and prognosis and recurrence prediction.

### 3.1. Circulating Tumor DNA

cfDNA is DNA released into body fluids due to cell apoptosis or necrosis, whereas ctDNA is DNA derived from tumor cells [33]. ctDNA is primarily fragmented DNA with <200 base pairs and a short half-life of 1–2 h, which is assumed to represent the tumor status in real time. ctDNA is frequently determined in <0.1% of cfDNA, and several analytical procedures have been established [38].

The most predominant methods for detecting ctDNA are digital polymerase chain reaction (dPCR) and next-generation sequencing (NGS). PCR is a process for denaturing target DNA, adding primers, and amplifying it. However, the amplified DNA is measured while in a plateau state; thus, variations in reaction speed cause differences in the results. Hence, the DNA sample in dPCR is divided into hundreds of wells, with the reaction taking place in each well. The presence of amplification in each well is assessed following the response. Wells with amplification are counted as positive, whereas those without amplification are counted as negative, indicating the absence of target DNA [39]. dPCR technologies involve droplet dPCR (ddPCR) and beads, emulsion, amplification, and magnetics (BEAMing). ddPCR splits DNA samples with oil droplets rather than microfluidic plates or chips, and it detects genomic products as little as 0.01% [33,40]. BEAMing is a technology that combines dPCR with flow cytometry [41]. This approach detects ctDNA with a similar sensitivity to ddPCR [33,41].

NGS simultaneously sequences thousands or millions of DNA and RNA molecules [42]. The first step is generating a DNA library by randomly adding adapter sequences to the DNA fragments. DNA linkers are hybridized with adapters to amplify each DNA fragment [43]. Several approaches have been established for NGS assays. Cancer personalized profiling by deep sequencing (CAPP-Seq) detects and quantifies genetic mutations by hybridizing a biotinylated oligonucleotide, “a selector”, to a target gene [44]. Tagged-amplicon deep sequencing (Tam-Seq) is a type of amplicon analysis that involves PCR-amplified target genes [45]. In this approach, the target gene is amplified with a single-plex reaction, the adapter and barcode sequences are then added to the amplicon, and PCR is repeated before sequencing [45]. Targeted sequencing, including CAPP-seq and Tam-seq, is relatively affordable and requires a short turnaround time (TAT). However, they require identifying the area of interest. The cost of NGS has decreased in recent years; thus, more thorough analyses, such as whole-exome sequencing and whole-genome sequencing, are becoming popular.

Several studies have been conducted in EC because ctDNA is relatively simple to isolate. Our group investigated postoperative ctDNA from ESCC with panel sequencing and revealed that ctDNA was related to the pathological response to NAC [46]. Patients who failed NAC demonstrated a greater ctDNA positivity rate than those who responded well. Furthermore, the postoperative ctDNA-positive group exhibited worse recurrence-free survival (RFS) than the negative group (1-year RFS rate: 0% vs. 90%, HR: 16.9, 95% CI: 1.92–149.4, *p* = 0.0008) [46]. This was consistent with EAC, where Ococks evaluated postoperative ctDNA and revealed that 90% of ctDNA-positive patients relapsed postoperatively [47]. Cancer-specific survival was substantially worse in the ctDNA-positive group than in the negative group [47]. Azad investigated ctDNA with CAPP-seq in patients with EC treated with CRT and revealed that the ctDNA-positive group demonstrated a worse prognosis [48]. He also reported a median ctDNA fraction of 0.07%, concluding that ctDNA analysis requires a very sensitive testing technology [48]. Chen conducted CRT with toripalimab, an anti-PD-1 antibody, on patients with ESCC and assessed ctDNA during and after the treatment [49]. The results indicated that the ctDNA-negative group exhibited a significantly higher clinical complete response (CR) rate. Additionally, the ctDNA-positive group demonstrated a worse prognosis than the ctDNA-negative group [49].

Several studies have demonstrated that ctDNA helps identify treatment efficacy and prognosis in EC, but numerous issues have been raised, including the possibility of false negatives in cases with low tumor burden and false positives because of clonal hematopoiesis of indeterminate potential (CHIP) [50,51]. A meta-analysis reported that the sensitivity and specificity of ctDNA for diagnosing EC were 71.0% and 98.6%, respectively [52]. However, regarding ctDNA in the case of low tumor burden, Iwaya reported that the positive rate of ctDNA was 14.3% in stage I ESCC, while it was 85.2% in stage II or higher [53]. Multiple studies have addressed these concerns with a tumor-informed approach, employing validation to match ctDNA in the blood with DNA from the tumor tissue. This method will generate more sensitive results because it utilizes liquid biopsy to target the mutations that are already found in tissue DNA. Signatera, which is the first assay to adopt the tumor-informed approach, has already been used for various cancers. In particular, Reinert’s study of postoperative ctDNA in colorectal cancer indicated that the sensitivity and specificity for recurrence were 88% and 98%, respectively, indicating remarkably high values [54]. Furthermore, Kasi claimed that ctDNA is modified by methylation signals in the blood and emphasized the use of a tumor-informed approach [55]. The tumor-informed approach has the disadvantage of longer TAT compared with blood analysis alone, but it may be highly effective, and more evidence is expected in the future [42].

### 3.2. CTC

CTC is a cancer cell that enters the bloodstream after passing through the basement membrane from the epithelium [56]. CTCs are hypothesized to contribute to cancer metastasis [57]. CTCs in the blood indicate that the cancer has advanced. They are present in very small numbers, ranging from a few to tens per milliliter of blood; thus, it is crucial in analysis to isolate CTCs from other blood cells [58]. Immunomagnetic and physical techniques are two primary approaches for detecting CTCs.

The immunomagnetic technique is predominantly used to isolate CTCs. This technique takes advantage of the immunological properties of CTCs and blood cells. CellSearch [59], MagSweeper [60], AdnaTest [61], and CTC-chip [62] are well-known systems, with CellSearch being the only Food and Drug Administration-approved device for CTC analysis. CellSearch detects CTCs by capturing a marker known as epithelial cell adhesion molecule (EpCAM), which is found in epithelial cells, utilizing an antigen–antibody reaction with magnetic beads covered in antibodies [59]. AdnaTest aims to enhance sensitivity using antibodies against epithelial cell markers other than EpCAM [61], whereas CTC-chip utilizes microposts instead of magnetic beads [62]. These immunomagnetic techniques have helped predict the prognosis of some malignancies. However, cancer cells that lose EpCAM expression due to epithelial–mesenchymal transition (EMT) through invasion into the blood are difficult to identify as CTCs [63]. Systems with an antibody against CD45, a leukocyte surface antigen, have been developed in recent years to isolate CTCs indirectly (negative selection) to address this issue [64].

Physical approaches take advantage of CTCs being bigger, denser, and more rigid than blood cells [58]. In particular, isolation by size of epithelial tumor cells (ISET) [65] and ScreenCell [66] collect CTCs with membrane filters. Ficoll and OncoQuick utilize density gradient centrifugation to isolate CTCs from blood cells [67]. These approaches are simpler than the immunomagnetic techniques, and they isolate CTCs after EMT without relying on cell surface markers. However, comparatively small CTCs cannot be recovered. Recently, systems that combine immunomagnetic and physical techniques, such as CTC-ichip, have been invented [68]. The tumor biology of CTCs can be investigated by developing CTC-derived cell lines in this procedure, as we can collect live cells [69,70].

Several studies have been published on CTCs in EC, and CTC has been demonstrated to help assess treatment efficacy and prognosis. Tanaka examined CTCs before and after chemotherapy and chemoradiotherapy in 38 patients with primary or recurrent advanced EC [71]. Patients with fewer CTCs demonstrated a greater rate of complete and partial responses than those with unchanged or increasing CTCs posttreatment [71]. The group exhibiting <2 CTCs before and after treatment showed a better OS than those with >2 CTCs [71]. Additionally, Li compared pretreatment CTCs in 129 patients with EC to those in 75 controls, setting 2 CTCs in peripheral blood of 3.2 mL as the threshold [72]. The diagnostic ability of the CTC for EC included 70.54% sensitivity and 96.74% specificity [72]. The group with >3 CTCs demonstrated a substantially worse prognosis than those with <3 CTCs [72]. Similar studies have demonstrated the value of CTC and it is anticipated to be a technique of liquid biopsy applied to various purposes, such as identifying prognostic factors and elucidating the cancer state.

The challenges of isolation are still major limitations of CTC. Cells that have undergone EMT cannot be detected in positive selection [63]. Unwanted cells can be included in negative selection because the analyzed cells are the remaining cells after removing cells expressing CD45, and it is difficult to determine whether they are tumor-derived cells or not [73]. It has also been pointed out that negative selection may remove a large number of CD45-positive blood cells, and this may result in rare CTCs being captured and lost in the massive flow of the blood cells [73]. Also, the physical approach can lose 20–50% to CTCs because CTCs are often similar in size to WBCs [73,74]. These limitations are issues that need to be addressed, and more research is required in the future.

### 3.3. miRNA

Liquid biopsy has mainly focused on ctDNA and CTC, which exist in trace levels in body fluids and have unstable traits [75]. Hence, the sensitivity is uncertain when the tumor burden is minimal. Recently, miRNAs have appeared as a promising biomarker [76]. MiRNAs are non-coding RNAs with 18–25 base pairs with a hairpin structure [77]. They may affect cell proliferation, differentiation, and apoptosis by controlling mRNA translation. The miRNA is initially transcribed from the miRNA gene by RNA polymerase II in the cell nucleus (pri-miRNA), which is then cleaved by enzymes, namely Drosha and Pasha, to generate a precursor pre-miRNA with a stem-loop structure of approximately 70 nucleotides [78,79]. Exportin 5 transports pre-miRNAs into the cytoplasm, where enzymes, such as RNAse III and Dicer, convert them into miRNAs [80,81]. The generated miRNA is either integrated into exosomes or microvesicles as an RNA-induced silencing complex or released extracellularly after binding to Argonaute protein 2 or high-density lipoprotein [82,83,84,85]. These structures protect miRNAs from nuclease and protease degradation and keep their stability in body fluids [77]. MiRNAs have been investigated using microarrays or quantitative PCR (qPCR), and dPCR and NGS have gained popularity in recent years [86,87,88,89]. MiRNA analysis requires reverse transcriptase to produce complementary DNA (cDNA), unlike ctDNA. However, a method without reverse transcriptase has just been developed [90].

Some studies have focused on miRNAs in the field of EC, but not as many as those on ctDNA and CTC. Our group investigated urine miRNAs in 10 patients with advanced ESCC, 20 with superficial ESCC, and 20 healthy individuals [91]. Consequently, 18 of the 1205 analyzed miRNAs revealed alterations that are consistent with tumor volume changes. Additionally, 3 of the 18 miRNAs (hsa-miR-4323, hsa-miR-6824-3p, and hsa-miR-6831-5p) were related to recurrence [91]. Li assessed miRNAs in 36 patients with ESCC and 36 healthy individuals and revealed that six miRNA types (miR-1972, miR-4274, miR-4701-3p, miR-6126, miR-1268a, and miR-4505) were altered based on the tumor status [92]. They subsequently developed a qPCR panel using these six miRNAs and validated its diagnostic capacity. Three cohorts (N = 342, 207, 226) were validated, with sensitivity and specificity of 92.00% and 89.17%, 90.32% and 91.04%, 91.07% and 88.07%, respectively. Additionally, a scoring system was established according to the presence of the six types of miRNAs, and a cut-off value was identified to compare progression-free survival (PFS) and OS. Among all three cohorts, the group with more miRNA alterations (higher score) demonstrated poorer PFS and OS [92]. This indicates that miRNAs help in stratifying EC cases.

While several studies show miRNA can be utilized as a target for liquid biopsy, there are some limitations. To begin, miRNAs are abundant in various body fluids such as blood, urine, and saliva, complicating the isolation of tumor-derived miRNAs [76]. Secondly, there is no standardized methodology for isolating and analyzing miRNAs in body fluids [76,93]. Therefore, a reliable approach for selectively evaluating tumor-derived miRNAs is needed.

## 4. Future Perspective

Liquid biopsy can be used in EC to evaluate treatment efficacy, prognosis, and recurrence. Novel concepts for treating EC have evolved in recent years, including esophageal preservation, adjuvant immune checkpoint inhibitor (ICI) therapy, and conversion surgery. The combination of liquid biopsy with these concepts will cause a more effective EC treatment (Figure 2).

### 4.1. Organ Preservation Approach

The response rate for the preoperative therapy of EC has dramatically improved. More cases are achieving pCR with preoperative treatment alone because of chemotherapy advancements. JCOG1109 revealed that the pCR rates for CF, DCF, and CF + RT were 2.0%, 19.8%, and 38.5%, respectively [24]. The pCR rates in the JCOG1804E trial, which is a feasibility study that assessed the efficacy and safety of nivolumab in addition to standard NAC regimens, were extremely high at 33.3% for DCF + nivolumab and 41.7% for FLOT + nivolumab [28,29]. However, the morbidity and mortality rates for esophagectomy remain high despite advances in surgical methods. An international multicenter study of 24 high-volume institutions in 14 countries revealed a 59% total morbidity rate, with a 30-day mortality rate of 2.4% [94]. Hence, determining pCR preoperatively and preserving the esophagus will remarkably benefit the patient.

Two primary esophageal preservation strategies exist, including definitive CRT (dCRT) for advanced cancer and active surveillance after NACRT. The JCOG0909 trial is a single-arm, non-randomized study of dCRT efficacy and safety (cisplatin of 75 mg/m^2^ and 5-FU of 1000 mg/m^2^ + 50.4 Gy) in 94 patients with cStage II/III ESCC [95]. Patients were assessed for response after dCRT, and those with a CR or partial response received up to two additional CF therapy courses. They proceeded to salvage surgery if the response indicated stable disease or progressive disease. The study revealed a 3-year survival rate of 74.2% (90% CI: 65.9–80.8) and an esophagectomy-free survival (EFS) rate of 63.6% (95% CI: 52.9–72.4). After dCRT, 21 patients underwent salvage esophagectomy, including two with pCR [95]. A randomized controlled trial (NEEDS trial) is currently ongoing in Europe, comparing surgery after dCRT (platin-taxane or platin-fluoropyrimidine + 50 or 50.4 Gy) to the CROSS regimen [96].

The CROC study is a trial that used dCRT (cisplatin of 75 mg/m^2^ and 5-FU of 1000 mg/m^2^ +50.4 Gy) in patients with ESCC who have significantly responded to preoperative DCF therapy instead of surgery [97]. After DCF, 58.4% of the patients demonstrated a remarkable response, and 89.8% achieved CR with dCRT. The 1-year PFS for the remarkable response and dCRT groups was 89.8% (95% CI: 77.2–95.6). The 3-year survival rate for all patients was 74.1% (95% CI: 62.2–82.8), and the EFS was 45.3% (95% CI: 34.4–55.6) [97]. This indicates that chemoselection, which stratifies individuals expected to be treated completely with dCRT based on their chemotherapy response, is promising.

The SANO trial is a study of active surveillance after NACRT [98]. In this trial, patients who achieved clinical CR (cCR) with the CROSS regimen were randomly assigned to active surveillance or surgery. Some results were released in 2023, revealing that active surveillance was not inferior to surgery in terms of survival (HR: 0.88, 95% upper limit: 1.40, *p* = 0.004) [99]. Furthermore, 48% of the patients under active surveillance exhibited locoregional recurrence, with 17% having distant metastases [99]. A similar trial, called ESOSTRATE, is ongoing in France aside from the SANO trial [100].

All of the above-listed esophageal preservation studies used conventional methods to evaluate the efficacy of perioperative therapy. They contribute to some degree of stratification, but they remain unsatisfactory. In particular, esophagectomy was performed on patients who did not react to chemotherapy after dCRT, including two with pCR, in the JCOG 0909 trial [95]. Furthermore, only 35% of patients on active surveillance in the SANO trial maintained cCR, raising questions about the value of these assessments [99]. Hence, liquid biopsy in these investigations may generate a more precise efficacy estimate. Liquid biopsy demonstrated a risk of being less sensitive when the tumor burden is low, but endoscopic imaging that incorporates artificial intelligence along with liquid biopsy has been proposed to compensate for this limitation [101,102].

### 4.2. Individualized Adjuvant ICI Therapy

The CheckMate 577 trial evaluated the efficacy and safety of nivolumab as an adjuvant therapy in patients with EC or esophagogastric junction cancer who previously underwent NACRT and surgery without pCR [25]. The study indicated that nivolumab improved prognosis, making adjuvant nivolumab an acceptable therapy option. However, the indication for adjuvant nivolumab is debatable, as no studies have demonstrated the efficacy of postoperative nivolumab in individuals who underwent surgery following NAC. Further, the side effects of nivolumab cannot be ignored. Nivolumab exhibited a severe adverse event rate of 8% (compared to 3% for the placebo) and a treatment discontinuation rate of 9% [25]. This adverse event profile is notable because immune-related adverse events from ICIs, such as nivolumab, can be permanent, unlike other medications [103]. Furthermore, the high cost of nivolumab raises problems in health economics in many countries with universal health insurance [104]. Therefore, the indication for adjuvant nivolumab should be carefully considered. Postoperative MRD can be assessed, which may provide a rationale for intensive postoperative therapy in MRD-positive cases if liquid biopsy can be performed in such circumstances.

### 4.3. Selecting the Appropriate Candidates for Conversion Surgery

Conversion surgery is a surgical treatment performed on patients who were initially deemed unresectable due to oncologic or technical reasons but are now resectable after primary therapy with downstaging [105]. Conversion surgery is becoming predominant as multidisciplinary treatment progresses, and numerous studies have established its efficacy. Makino conducted a systematic review of conversion surgery for cT4 EC and revealed that conversion esophagectomy improved the prognosis [106]. Additionally, Tsuji studied conversion surgery for advanced EC with distant metastases and revealed that it was associated with a longer prognosis in patients with a pathologic response (HR: 0.493, 95% CI: 0.283–0.859, *p* = 0.012) [107]. This study emphasizes the importance of predicting the responder before undergoing conversion surgery. However, no explicit standards exist for conversion surgery timing. In practice, surgeons make subjective decisions on whether R0 resection is oncologically and technically possible based on imaging results from endoscopy, CT, and PET. Hence, the tumor volume throughout the body can be accurately assessed and a more objective decision can be made if liquid biopsy is used as a supplement to conventional methods to determine the timing of conversion surgery.

## 5. Conclusions

Advances in multidisciplinary treatment have led to dramatic changes in the management of EC. Liquid biopsy research is advancing for various agents, including ctDNA, CTC, and miRNA, and is expected to become more widespread with the introduction of new technologies, including NGS. The integration of liquid biopsy with conventional modalities such as endoscopy and CT scan during the treatment process allows for a more accurate assessment of tumor volume, leading to the development of new concepts such as organ preservation, adjuvant ICI therapy, and conversion surgery. Their potential to improve treatment decisions will further advance the management of EC.

## Figures and Tables

**Figure 1 cancers-17-00196-f001:**
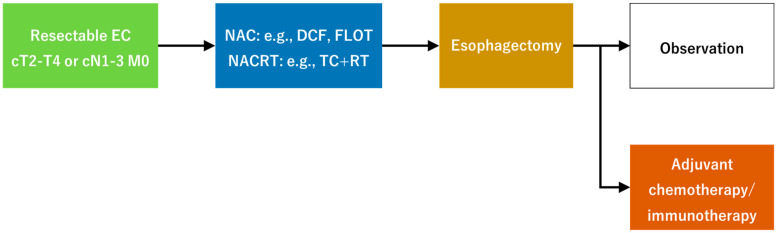
Treatment for resectable esophageal cancer. EC, esophageal cancer; NAC, neoadjuvant chemotherapy; NACRT, neoadjuvant chemoradiotherapy; TC+RT, paclitaxel + carboplatin + radiotherapy.

**Figure 2 cancers-17-00196-f002:**
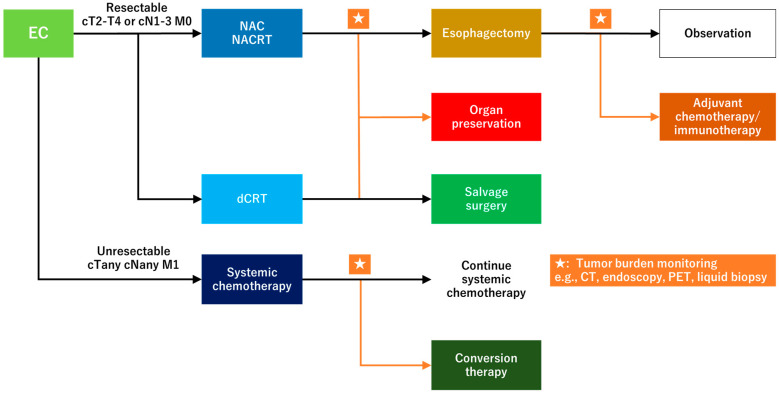
Future perspective of combining liquid biopsy with esophageal cancer treatment. EC, esophageal cancer; NAC, neoadjuvant chemotherapy; NACRT, neoadjuvant chemoradiotherapy; dCRT, definitive chemoradiotherapy.

## Data Availability

No new data were created or analyzed in this study. Data sharing is not applicable to this article.

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
