# Peer review of "Liquid Biopsy and Multidisciplinary Treatment for Esophageal Cancer"

_cancers, 2025, doi:10.3390/cancers17020196_

Round 1

Reviewer 1 Report

Comments and Suggestions for Authors

The manuscript entitled "[Liquid Biopsy and Multidisciplinary Treatment for Esophageal Cancer]” provides a valuable review of liquid biopsy technologies (ctDNA, CTC, miRNA) in the treatment of esophageal cancer, highlighting their potential in treatment monitoring, prognosis, and recurrence. It effectively fills a research gap and discusses the integration of liquid biopsy into clinical workflows.

However, I recommend the following improvements:

  • Discussion of more relevant data and the inclusion of specific new clinical data (e.g., ctDNA, CTCs, miRNA) would enhance the credibility and impact of the manuscript.
  • Furthermore, although the need for and use of non-invasive new technologies such as ctDNA, miRNA, or CTCs (e.g., NGS, dPCR) have been mentioned, it would be appropriate to provide a more detailed discussion of the differences, advantages, and disadvantages of these methods.
  • Expanding the discussion on the limitations of liquid biopsy technologies in scenarios such as low tumor burden or minimal disease would make a significant contribution to the manuscript.
  • Overall, the figures (Figure 1 and Figure 2) are not very informative. Both figures on clinical applications of liquid biopsy technologies should be made more detailed and visually appealing to engage the reader.
  • The conclusions drawn from the evaluated findings should be strengthened by the authors clearly expressing their perspectives. This would significantly contribute to a more in-depth discussion and effective emphasis in the conclusion.
  • Lastly, the manuscript should undergo a comprehensive grammatical review and correction of any spelling mistakes.
  • Overall, the manuscript makes a valuable contribution to the field. Addressing these recommendations would provide a more comprehensive and balanced perspective on the clinical benefits and challenges of liquid biopsy technologies in the treatment of esophageal cancer.

Reviewer 2 Report

Comments and Suggestions for Authors

Dear Editor and Authors,

Thank you for asking me to review this manuscript titled “Liquid Biopsy and Multidisciplinary Treatment for Esophageal Cancer” by Dr. Hoshi and his colleagues from the Department of Surgery, Keio University School of Medicine in Tokyo, Japan.

In this review work the authors present a very extensive and comprehensive overview of the use of liquid biopsy as a method of they suggest monitoring disease progression and response. It is a relatively well written work which only requires some minor language editing. Its structure and flow is appropriate and the information quite elaborate and complete. At certain point’s the authors become quite exuberant and carried away in their claims! As a work it is a good presentation of the whole notion of liquid biopsy and the literature - evidence available. It does have a certain degree of merit and could prove useful to clinicians and researchers alike in summarizing and collecting together appropriate evidence. I only have a few minor comments to make:

Comments:

1.       The authors talk about using liquid biopsy as a means of evaluating and monitoring disease response/progression and mention it can detect “minimal disease” but how accurate is this?? What is the tumor burden one must have for liquid biopsy to be applicable? What is the sensitivity of the method? These need addressing and clarifying!

2.       Liquid biopsy still has significant drawbacks, it certainly needs refinement and even the authors acknowledge this in their work. However, I feel they need to taper down their enthusiasm about this method for it quite frankly deflate quickly as a punctured balloon! It does seem to have potential and this is what the level of the work should be at and not a full hearted support shown!!

3.       I am not sure what sections 4.1, 4.2 and 4.3 aim to! They discuss management outside the scope of liquid biopsy in EC and should probably be removed!

Minor Comments:

1.       Fix line 82 – say these are trials!
